# Optimized Pre-Processing for Discrimination Prevention

**Flavio P. Calmon**
Harvard University
flavio@seas.harvard.edu

**Dennis Wei**
IBM Research AI
dwei@us.ibm.com

**Bhanukiran Vinzamuri**
IBM Research AI
bhanu.vinzamuri@ibm.com

**Karthikeyan Natesan Ramamurthy**
IBM Research AI
knatesa@us.ibm.com

**Kush R. Varshney**
IBM Research AI
krvarshn@us.ibm.com

## Abstract

Non-discrimination is a recognized objective in algorithmic decision making. In this paper, we introduce a novel probabilistic formulation of data pre-processing for reducing discrimination. We propose a convex optimization for learning a data transformation with three goals: controlling discrimination, limiting distortion in individual data samples, and preserving utility. We characterize the impact of limited sample size in accomplishing this objective. Two instances of the proposed optimization are applied to datasets, including one on real-world criminal recidivism. Results show that discrimination can be greatly reduced at a small cost in classification accuracy.

## 1 Introduction

Discrimination is the prejudicial treatment of an individual based on membership in a legally protected group such as a race or gender. Direct discrimination occurs when protected attributes are used explicitly in making decisions, also known as *disparate treatment*. More pervasive nowadays is indirect discrimination, in which protected attributes are not used but reliance on variables correlated with them leads to significantly different outcomes for different groups. The latter phenomenon is termed *disparate impact*. Indirect discrimination may be intentional, as in the historical practice of "redlining" in the U.S. in which home mortgages were denied in zip codes populated primarily by minorities. However, the doctrine of disparate impact applies regardless of actual intent.

Supervised learning algorithms, increasingly used for decision making in applications of consequence, may at first be presumed to be fair and devoid of inherent bias, but in fact, inherit any bias or discrimination present in the data on which they are trained [Calders and Žliobaitė, 2013]. Furthermore, simply removing protected variables from the data is not enough since it does nothing to address indirect discrimination and may in fact conceal it. The need for more sophisticated tools has made discrimination discovery and prevention an important research area [Pedreschi et al., 2008].

Algorithmic discrimination prevention involves modifying one or more of the following to ensure that decisions made by supervised learning methods are less biased: (a) the training data, (b) the learning algorithm, and (c) the ensuing decisions themselves. These are respectively classified as pre-processing [Hajian, 2013], in-processing [Fish et al., 2016, Zafar et al., 2016, Kamishima et al., 2011] and post-processing approaches [Hardt et al., 2016]. In this paper, we focus on pre-processing since it is the most flexible in terms of the data science pipeline: it is independent of the modeling algorithm and can be integrated with data release and publishing mechanisms.

Researchers have also studied several notions of discrimination and fairness. Disparate impact is addressed by the principles of *statistical parity* and *group fairness* [Feldman et al., 2015], which seek similar outcomes for all groups. In contrast, *individual fairness* [Dwork et al., 2012] mandates that similar individuals be treated similarly irrespective of group membership. For classifiers and other

predictive models, equal error rates for different groups are a desirable property [Hardt et al., 2016], as is calibration or lack of *predictive bias* in the predictions [Zhang and Neill, 2016]. The tension between the last two notions is described by Kleinberg et al. [2017] and Chouldechova [2016]; the work of Friedler et al. [2016] is in a similar vein. Corbett-Davies et al. [2017] discuss the trade-offs in satisfying prevailing notions of algorithmic fairness from a public safety standpoint. Since the present work pertains to pre-processing and not modeling, balanced error rates and predictive bias are less relevant criteria. Instead we focus primarily on achieving group fairness while also accounting for individual fairness through a distortion constraint.

Existing pre-processing approaches include sampling or re-weighting the data to neutralize discriminatory effects [Kamiran and Calders, 2012], changing the individual data records [Hajian and Domingo-Ferrer, 2013], and using $t$-closeness [Li et al., 2007] for discrimination control [Ruggieri, 2014]. A common theme is the importance of balancing discrimination control against utility of the processed data. However, this prior work neither presents general and principled optimization frameworks for trading off these two criteria, nor allows connections to be made to the broader statistical learning and information theory literature via probabilistic descriptions. Another shortcoming is that individual distortion or fairness is not made explicit.

In this work, we (i) introduce a probabilistic framework for discrimination-preventing pre-processing in supervised learning, (ii) formulate an optimization problem for producing pre-processing transformations that trade off discrimination control, data utility, and individual distortion, (iii) characterize theoretical properties of the optimization approach (e.g. convexity, robustness to limited samples), and (iv) benchmark the ensuing pre-processing transformations on real-word datasets. Our aim in part is to work toward a more unified view of existing

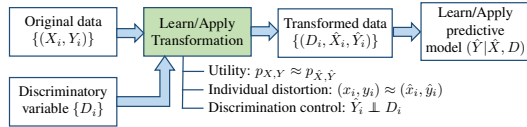

Figure 1: The proposed pipeline for predictive learning with discrimination prevention. *Learn* mode applies with training data and *apply* mode with novel test data. Note that test data also requires transformation before predictions can be obtained.

pre-processing concepts and methods, which may help to suggest refinements. While discrimination and utility are defined at the level of probability distributions, distortion is controlled on a per-sample basis, thereby limiting the effect of the transformation on individuals and ensuring a degree of individual fairness. Figure 1 illustrates the supervised learning pipeline that includes our proposed discrimination-preventing pre-processing.

The work of Zemel et al. [2013] is closest to ours in also presenting a framework with three criteria related to discrimination control (group fairness), individual fairness, and utility. However, the criteria are manifested less directly than in our proposal. Discrimination control is posed in terms of intermediate features rather than outcomes, individual distortion does not take outcomes into account (being an $\ell_2$-norm between original and transformed features), and utility is specific to a particular classifier. Our formulation more naturally and generally encodes these fairness and utility desiderata.

Given the novelty of our formulation, we devote more effort than usual to discussing its motivations and potential variations. We state conditions under which the proposed optimization problem is convex. The optimization assumes as input an estimate of the distribution of the data which, in practice, can be imprecise due to limited sample size. Accordingly, we characterize the possible degradation in discrimination and utility guarantees at test time in terms of the training sample size. To demonstrate our framework, we apply specific instances of it to a prison recidivism dataset [ProPublica, 2017] and the UCI Adult dataset [Lichman, 2013]. We show that discrimination, distortion, and utility loss can be controlled simultaneously with real data. We also show that the pre-processed data reduces discrimination when training standard classifiers, particularly when compared to the original data with and without removing protected variables. In the Supplementary Material (SM), we describe in more detail the resulting transformations and the demographic patterns that they reveal.

## 2 General Formulation

We are given a dataset consisting of $n$ i.i.d. samples $\{(D_i, X_i, Y_i)\}_{i=1}^n$ from a joint distribution $p_{D,X,Y}$ with domain $\mathcal{D} \times \mathcal{X} \times \mathcal{Y}$. Here $D$ denotes one or more protected (discriminatory) variables such as gender and race, $X$ denotes other non-protected variables used for decision making, and $Y$ is an *outcome* random variable. We use the term 'discriminatory' interchangeably with 'protected,'

and not in the usual statistical sense. For instance, $Y_i$ could represent a loan approval decision for individual $i$ based on demographic information $D_i$ and credit score $X_i$. We focus in this paper on discrete (or discretized) and finite domains $\mathcal{D}$ and $\mathcal{X}$ and binary outcomes, i.e. $\mathcal{Y} = \{0, 1\}$. There is no restriction on the dimensions of $D$ and $X$.

Our goal is to determine a randomized mapping $p_{\hat{X}, \hat{Y}|X,Y,D}$ that (i) transforms the given dataset into a new dataset $\{(D_i, \hat{X}_i, \hat{Y}_i)\}_{i=1}^n$ which may be used to train a model, and (ii) similarly transforms data to which the model is applied, i.e. test data. Each $(\hat{X}_i, \hat{Y}_i)$ is drawn independently from the same domain $\mathcal{X} \times \mathcal{Y}$ as $X, Y$ by applying $p_{\hat{X}, \hat{Y}|X,Y,D}$ to the corresponding triplet $(D_i, X_i, Y_i)$. Since $D_i$ is retained as-is, we do not include it in the mapping to be determined. Motivation for retaining $D$ is discussed later in Section 3. For test samples, $Y_i$ is not available at the input while $\hat{Y}_i$ may not be needed at the output. In this case, a reduced mapping $p_{\hat{X}|X,D}$ is used as given later in (9).

It is assumed that $p_{D,X,Y}$ is known along with its marginals and conditionals. This assumption is often satisfied using the empirical distribution of $\{(D_i, X_i, Y_i)\}_{i=1}^n$. In Section 3, we state a result ensuring that discrimination and utility loss continue to be controlled if the distribution used to determine $p_{\hat{X}, \hat{Y}|X,Y,D}$ differs from the distribution of test samples.

We propose that the mapping $p_{\hat{X}, \hat{Y}|X,Y,D}$ satisfy the three following properties.

**I. Discrimination Control.** The first objective is to limit the dependence of the transformed outcome $\hat{Y}$ on the protected variables $D$. We propose two alternative formulations. The first requires the conditional distribution $p_{\hat{Y}|D}$ to be close to a target distribution $p_{Y_T}$ for all values of $D$,

$$J\left(p_{\hat{Y}|D}(y|d), p_{Y_T}(y)\right) \leq \epsilon_{y,d} \ \forall \ d \in \mathcal{D}, y \in \{0, 1\}, \tag{1}$$

where $J(\cdot, \cdot)$ denotes some distance function. In the second formulation, we constrain the conditional probability $p_{\hat{Y}|D}$ to be similar for any two values of $D$:

$$J\left(p_{\hat{Y}|D}(y|d_1), p_{\hat{Y}|D}(y|d_2)\right) \leq \epsilon_{y,d_1,d_2} \ \forall \ d_1, d_2 \in \mathcal{D}, y \in \{0, 1\}. \tag{2}$$

Note that the number of such constraints is $O(|\mathcal{D}|^2)$ as opposed to $O(|\mathcal{D}|)$ constraints in (1). The choice of $p_{Y_T}$ in (1), and $J$ and $\epsilon$ in (1) and (2) should be informed by societal aspects, consultations with domain experts and stakeholders, and legal considerations such as the "80% rule" [EEOC, 1979].

For this work, we choose $J$ to be the following probability ratio measure:

$$J(p, q) = \left| \frac{p}{q} - 1 \right|. \tag{3}$$

This metric is motivated by the "80% rule." The combination of (3) and (1) generalizes the extended lift criterion proposed in the literature [Pedreschi et al., 2012], while the combination of (3) and (2) generalizes selective and contrastive lift. The latter combination (2), (3) is used in the numerical results in Section 4. We note that the selection of a 'fair' target distribution $p_{Y_T}$ in (1) is not straightforward; see Žliobaitė et al. [2011] for one such proposal. Despite its practical motivation, we alert the reader that (3) may be unnecessarily restrictive when $q$ is low.

In (1) and (2), discrimination control is imposed jointly with respect to all protected variables, e.g. all combinations of gender and race if $D$ consists of those two variables. An alternative is to take the protected variables one at a time, and impose univariate discrimination control. In this work, we opt for the more stringent joint discrimination control, although legal formulations tend to be of the univariate type.

Formulations (1) and (2) control discrimination at the level of the overall population in the dataset. It is also possible to control discrimination within segments of the population by conditioning on additional variables $B$, where $B$ is a subset of $X$ and $X$ is a collection of features. Constraint (1) would then generalize to $J\left(p_{\hat{Y}|D,B}(y|d,b), p_{Y_T|B}(y|b)\right) \leq \epsilon_{y,d,b}$ for all $d \in \mathcal{D}$, $y \in \{0, 1\}$, and $b \in \mathcal{B}$. Similar conditioning or 'context' for discrimination has been explored before in Hajian and Domingo-Ferrer [2013] in the setting of association rule mining. For example, $B$ could represent the fraction of a pool of applicants that applied to a certain department, which enables the metric to avoid statistical traps such as the Simpson's paradox [Pearl, 2014]. One may wish to control for such

variables in determining the presence of discrimination, while ensuring that population segments created by conditioning are large enough to derive statistically valid inferences. Moreover, we note that there may exist inaccessible latent variables that drive discrimination, and the metrics used here are inherently limited by the available data. Recent definitions of fairness that seek to mitigate this issue include [Johnson et al., 2016] and [Kusner et al., 2017]. We defer further investigation of causality and conditional discrimination to future work.

**II. Distortion Control.** The mapping $p_{\hat{X},\hat{Y}|X,Y,D}$ should satisfy distortion constraints with respect to the domain $\mathcal{X} \times \mathcal{Y}$. These constraints restrict the mapping to reduce or avoid altogether certain large changes (e.g. a very low credit score being mapped to a very high credit score). Given a distortion metric $\delta : (\mathcal{X} \times \mathcal{Y})^2 \to \mathbb{R}_+$, we constrain the conditional expectation of the distortion as,

$$\mathbb{E}\left[\delta((x,y),(\hat{X},\hat{Y})) \mid D = d, X = x, Y = y\right] \le c_{d,x,y} \ \forall \ (d,x,y) \in \mathcal{D} \times \mathcal{X} \times \mathcal{Y}. \tag{4}$$

We assume that $\delta(x,y,x,y) = 0$ for all $(x,y) \in \mathcal{X} \times \mathcal{Y}$. Constraint (4) is formulated with pointwise conditioning on $(D,X,Y) = (d,x,y)$ in order to promote *individual* fairness. It ensures that distortion is controlled for every combination of $(d,x,y)$, i.e. every individual in the original dataset, and more importantly, every individual to which a model is later applied. By way of contrast, an average-case measure in which an expectation is also taken over $D, X, Y$ may result in high distortion for certain $(d,x,y)$, likely those with low probability. Equation (4) also allows the level of control $c_{d,x,y}$ to depend on $(d,x,y)$ if desired. We also note that (4) is a property of the mapping $p_{\hat{X},\hat{Y}|D,X,Y}$, and does not depend on the assumed distribution $p_{D,X,Y}$.

The expectation over $\hat{X}, \hat{Y}$ in (4) encompasses several cases depending on the choices of the metric $\delta$ and thresholds $c_{d,x,y}$. If $c_{d,x,y} = 0$, then no mappings with nonzero distortion are allowed for individuals with original values $(d,x,y)$. If $c_{d,x,y} > 0$, then certain mappings may still be disallowed by assigning them infinite distortion. Mappings with finite distortion are permissible subject to the budget $c_{d,x,y}$. Lastly, if $\delta$ is binary-valued (perhaps achieved by thresholding a multi-valued distortion function), it can be seen as classifying mappings into desirable ($\delta = 0$) and undesirable ones ($\delta = 1$). Here, (4) reduces to a bound on the conditional probability of an undesirable mapping, i.e.,

$$\Pr\left(\delta((x,y),(\hat{X},\hat{Y})) = 1 \mid D = d, X = x, Y = y\right) \le c_{d,x,y}. \tag{5}$$

**III. Utility Preservation.** In addition to constraints on individual distortions, we also require that the *distribution* of $(\hat{X}, \hat{Y})$ be statistically close to the distribution of $(X, Y)$. This is to ensure that a model learned from the transformed dataset (when averaged over the protected variables $D$) is not too different from one learned from the original dataset, e.g. a bank's existing policy for approving loans. For a given dissimilarity measure $\Delta$ between probability distributions (e.g. KL-divergence), we require that $\Delta\left(p_{\hat{X},\hat{Y}}, p_{X,Y}\right)$ be small.

**Optimization Formulation.** Putting together the considerations from the three previous subsections, we arrive at the optimization problem below for determining a randomized transformation $p_{\hat{X},\hat{Y}|X,Y,D}$ mapping each sample $(D_i, X_i, Y_i)$ to $(\hat{X}_i, \hat{Y}_i)$:

$$\min_{p_{\hat{X},\hat{Y}|X,Y,D}} \ \Delta\left(p_{\hat{X},\hat{Y}}, p_{X,Y}\right)$$
$$\text{s.t.} \ J\left(p_{\hat{Y}|D}(y|d), p_{Y_T}(y)\right) \le \epsilon_{y,d} \ \text{and}$$
$$\mathbb{E}\left[\delta((x,y),(\hat{X},\hat{Y})) \mid D = d, X = x, Y = y\right] \le c_{d,x,y} \ \forall \ (d,x,y) \in \mathcal{D} \times \mathcal{X} \times \mathcal{Y},$$
$$p_{\hat{X},\hat{Y}|X,Y,D} \ \text{is a valid distribution.} \tag{6}$$

We choose to minimize the utility loss $\Delta$ subject to constraints on individual distortion (4) and discrimination (we use (1) for concreteness, but (2) can be used instead), since it is more natural to place bounds on the latter two.

The distortion constraints (4) are an essential component of the problem formulation (6). Without (4) and assuming that $p_{Y_T} = p_Y$, it is possible to achieve perfect utility and non-discrimination simply by sampling $(\hat{X}_i, \hat{Y}_i)$ from the original distribution $p_{X,Y}$ independently of any inputs, i.e.

$p_{\hat{X},\hat{Y}|X,Y,D}(\hat{x},\hat{y}|x,y,d) = p_{\hat{X},\hat{Y}}(\hat{x},\hat{y}) = p_{X,Y}(\hat{x},\hat{y})$. Then $\Delta(p_{\hat{X},\hat{Y}}, p_{X,Y}) = 0$, and $p_{\hat{Y}|D}(y|d) = p_{\hat{Y}}(y) = p_Y(y) = p_{Y_T}(y)$ for all $d \in \mathcal{D}$. Clearly, this solution is objectionable from the viewpoint of individual fairness, especially for individuals to whom a subsequent model is applied since it amounts to discarding an individual's data and replacing it with a random sample from the population $p_{X,Y}$. Constraint (4) seeks to prevent such gross deviations from occurring. The distortion constraints may, however, render the optimization infeasible, as illustrated in the SM.

## 3 Theoretical Properties

**I. Convexity.** We show conditions under which (6) is a convex or quasiconvex optimization problem, and can thus be solved to optimality. The proof is presented in the SM.

**Proposition 1.** *Problem* (6) *is a (quasi)convex optimization if* $\Delta(\cdot,\cdot)$ *is (quasi)convex and* $J(\cdot,\cdot)$ *is quasiconvex in their respective first arguments (with the second arguments fixed). If discrimination constraint* (2) *is used in place of* (1)*, then the condition on* $J$ *is that it be jointly quasiconvex in both arguments.*

**II. Generalizability of Discrimination Control.** We now discuss the generalizability of discrimination guarantees (1) and (2) to unseen individuals, i.e. those to whom a model is applied. Recall from Section 2 that the proposed transformation retains the protected variables $D$. We first consider the case where models trained on the transformed data to predict $\hat{Y}$ are allowed to depend on $D$. While such models may qualify as disparate treatment, the intent and effect is to better mitigate disparate impact resulting from the model. In this respect our proposal shares the same spirit with 'fair' affirmative action in Dwork et al. [2012] (fairer on account of distortion constraint (4)).

Assuming that predictive models for $\hat{Y}$ can depend on $D$, let $\widetilde{Y}$ be the output of such a model based on $D$ and $\hat{X}$. To remove the separate issue of model accuracy, suppose for simplicity that the model provides a good approximation to the conditional distribution of $\hat{Y}$, i.e. $p_{\widetilde{Y}|\hat{X},D}(\widetilde{y}|\hat{x},d) \approx p_{\hat{Y}|\hat{X},D}(\widetilde{y}|\hat{x},d)$. Then for individuals in a protected group $D = d$, the conditional distribution of $\widetilde{Y}$ is given by

$$p_{\widetilde{Y}|D}(\widetilde{y}|d) = \sum_{\hat{x}} p_{\widetilde{Y}|\hat{X},D}(\widetilde{y}|\hat{x},d)p_{\hat{X}|D}(\hat{x}|d) \approx \sum_{\hat{x}} p_{\hat{Y}|\hat{X},D}(\widetilde{y}|\hat{x},d)p_{\hat{X}|D}(\hat{x}|d) = p_{\hat{Y}|D}(\widetilde{y}|d). \quad (7)$$

Hence the model output $p_{\widetilde{Y}|D}$ can also be controlled by (1) or (2).

On the other hand, if $D$ must be suppressed from the transformed data, perhaps to comply with legal requirements regarding its non-use, then a predictive model can depend only on $\hat{X}$ and approximate $p_{\hat{Y}|\hat{X}}$, i.e. $p_{\widetilde{Y}|\hat{X},D}(\widetilde{y}|\hat{x},d) = p_{\widetilde{Y}|\hat{X}}(\widetilde{y}|\hat{x}) \approx p_{\hat{Y}|\hat{X}}(\widetilde{y}|\hat{x})$. In this case we have

$$p_{\widetilde{Y}|D}(\widetilde{y}|d) \approx \sum_{\hat{x}} p_{\hat{Y}|\hat{X}}(\widetilde{y}|\hat{x})p_{\hat{X}|D}(\hat{x}|d), \quad (8)$$

which in general is not equal to $p_{\hat{Y}|D}(\widetilde{y}|d)$ in (7). The quantity on the right-hand side of (8) is less straightforward to control. We address this question in the SM.

**III. Training and Application Considerations.** The proposed optimization framework has two modes of operation (Fig. 1): train and apply. In train mode, the optimization problem (6) is solved in order to determine a mapping $p_{\hat{X},\hat{Y}|X,Y,D}$ for randomizing the training set. The randomized training set, in turn, is used to fit a classification model $f_\theta(\hat{X},D)$ that approximates $p_{\hat{Y}|\hat{X},D}$, where $\theta$ are the parameters of the model. At apply time, a new data point $(X,D)$ is received and transformed into $(\hat{X},D)$ through a randomized mapping $p_{\hat{X}|X,D}$. The mapping $p_{\hat{X}|D,X}$ is given by marginalizing over $Y,\hat{Y}$:

$$p_{\hat{X}|D,X}(\hat{x}|d,x) = \sum_{y,\hat{y}} p_{\hat{X},\hat{Y}|X,Y,D}(\hat{x},\hat{y}|x,y,d)p_{Y|X,D}(y|x,d). \quad (9)$$

Assuming that the variable $D$ is not suppressed, and that the marginals are known, then the utility and discrimination guarantees set during train time still hold during apply time, as discussed above.

However, the distortion control will inevitably change, since the mapping has been marginalized over $Y$. More specifically, the bound on the expected distortion for each sample becomes

$$\mathbb{E}\left[\mathbb{E}\left[\delta((x,Y),(\hat{X},\hat{Y}))\mid D=d, X=x, Y\right]\mid D=d, X=x\right] \leq \sum_{y\in\mathcal{Y}} p_{Y|X,D}(y|x,d)c_{x,y,d} \triangleq c_{x,d}\,.$$
(10)

If the distortion control values $c_{x,y,d}$ are independent of $y$, then the upper-bound on distortion set during training time still holds during apply time. Otherwise, (10) provides a bound on individual distortion at apply time. The same guarantee holds for the case when $D$ is suppressed.

**IV. Robustness to Mismatched Prior Distribution Estimation.** We may also consider the case where the distribution $p_{D,X,Y}$ used to determine the transformation differs from the distribution $q_{D,X,Y}$ of test samples. This occurs, for example, when $p_{D,X,Y}$ is the empirical distribution computed from $n$ i.i.d. samples from an unknown distribution $q_{D,X,Y}$. In this situation, discrimination control and utility are still guaranteed for samples drawn from $q_{D,X,Y}$ that are transformed using $p_{\hat{Y},\hat{X}|X,Y,D}$, where the latter is obtained by solving (6) with $p_{D,X,Y}$. In particular, denoting by $q_{\hat{Y}|D}$ and $q_{\hat{X},\hat{Y}}$ the corresponding distributions for $\hat{Y}, \hat{X}$ and $D$ when $q_{D,X,Y}$ is transformed using $p_{\hat{Y},\hat{X}|X,Y,D}$, we have $J\left(p_{\hat{Y}|D}(y|d), p_{Y_T}(y)\right) \rightarrow J\left(q_{\hat{Y}|D}(y|d), p_{Y_T}(y)\right)$ and $\Delta\left(p_{X,Y}, p_{\hat{X},\hat{Y}}\right) \rightarrow \Delta\left(q_{X,Y}, q_{\hat{X},\hat{Y}}\right)$ for $n$ sufficiently large (the distortion control constraints (4) only depend on $p_{\hat{Y},\hat{X}|X,Y,D}$). The next proposition provides an estimate of the rate of this convergence in terms of $n$ and assuming $p_{Y,D}(y,d)$ is fixed and bounded away from zero. Its proof can be found in the SM.

**Proposition 2.** *Let $p_{D,X,Y}$ be the empirical distribution obtained from $n$ i.i.d. samples that is used to determine the mapping $p_{\hat{Y},\hat{X}|X,Y,D}$, and $q_{D,X,Y}$ be the true distribution of the data, with support size $m \triangleq |\mathcal{X} \times \mathcal{Y} \times \mathcal{D}|$. In addition, denote by $q_{D,\hat{X},\hat{Y}}$ the joint distribution after applying $p_{\hat{Y},\hat{X}|X,Y,D}$ to samples from $q_{D,X,Y}$. If for all $y \in \mathcal{Y}$, $d \in \mathcal{D}$ we have $p_{Y,D}(y,d) > 0$, $J\left(p_{\hat{Y}|D}(y|d), p_{Y_T}(y)\right) \leq \epsilon$, where $J$ is given in (3), and*

$$\Delta\left(p_{X,Y}, p_{\hat{X},\hat{Y}}\right) = \sum_{x,y}\left|p_{X,Y}(x,y) - p_{\hat{X},\hat{Y}}(x,y)\right| \leq \mu,$$

*with probability $1 - \beta$,*

$$\max\left\{J\left(q_{\hat{Y}|D}(y|d), p_{Y_T}(y)\right) - \epsilon, \Delta\left(q_{X,Y}, q_{\hat{X},\hat{Y}}\right) - \mu\right\} \lesssim \sqrt{\frac{m}{n}\log\left(1 + \frac{n}{m}\right) - \frac{\log\beta}{n}}.$$
(11)

Proposition 2 guarantees that, as long as $n$ is sufficiently large, the utility and discrimination control guarantees will approximately hold when $p_{\hat{X},\hat{Y}|Y,X,D}$ is applied to fresh samples drawn from $q_{D,X,Y}$. In particular, the utility and discrimination guarantees will converge to the ones used as parameters in the optimization at a rate that is at least $\sqrt{\frac{1}{n}\log n}$. The distortion control guarantees (4) are a property of the mapping $p_{\hat{X},\hat{Y}|Y,X,D}$, and do not depend on the distribution of the data. The convergence rate is tied to the support size, and for large $m$ a dimensionality reduction step may be required to assuage generalization issues. The same upper bound on convergence rate holds for discrimination constraints of the form (2).

## 4 Experimental Results

This section provides a numerical demonstration of running the data processing pipeline in Fig. 1. Our focus here is on the discrimination-accuracy trade-off obtained when the pre-processed data is used to train standard prediction algorithms. The SM presents additional results on the trade-off between discrimination control $\epsilon$ and utility $\Delta$ as well as an analysis of the optimized data transformations.

We apply the pipeline to ProPublica's COMPAS recidivism data [ProPublica, 2017] and the UCI Adult dataset [Lichman, 2013]. From the COMPAS dataset (7214 instances), we select severity of charge, number of prior crimes, and age category to be the decision variables ($X$). The outcome variable ($Y$) is a binary indicator of whether the individual recidivated (re-offended), and race is set to be the protected variable ($D$). The encoding of categorical variables is described in the SM. For the Adult dataset (32561 instances), the features were categorized as protected variables ($D$):

gender (male, female); decision variables ($X$): age (quantized to decades) and education (quantized to years); and response variable ($Y$): income (binary).

Our proposed approach is benchmarked against two baselines, leaving the dataset as-is and suppressing the protected variable $D$ during training and testing. We also compare against the *learning fair representations* (LFR) algorithm from Zemel et al. [2013]. As discussed in the introduction, LFR has fundamental differences from the proposed framework. In particular, LFR only considers binary-valued $D$, and consequently, we restrict $D$ to be binary in the experiments presented here. However, our method is *not* restricted to $D$ being binary or univariate. Illustrations of our method on non-binary $D$ are provided in the SM.

The details of applying our method to the datasets are as follows. For each train/test split, we approximate $p_{D,X,Y}$ using the empirical distribution of $(D, X, Y)$ in the training set and solve (6) using a standard convex solver [Diamond and Boyd, 2016]. For both datasets the utility metric $\Delta$ is the total variation distance, i.e. $\Delta\left(p_{X,Y}, p_{\hat{X},\hat{Y}}\right) = \frac{1}{2}\sum_{x,y}\left|p_{X,Y}(x,y) - p_{\hat{X},\hat{Y}}(x,y)\right|$, the distortion constraint is the combination of (2) and (3), and two levels of discrimination control are used, $\epsilon = \{0.05, 0.1\}$. The distortion function $\delta$ is chosen differently for the two datasets as described below, based on the differing semantics of the variables in the two applications. The specific values were chosen for demonstration purposes to be reasonable to our judgment and can easily be tuned according to the desires of a practitioner. We emphasize that the distortion values were not selected to optimize the results presented here. All experiments run in minutes on a standard laptop.

*Distortion function for COMPAS:* We use the expected distortion constraint in (4) with $c_{d,x,y} = 0.4, 0.3$ for $d$ being respectively *African-American* and *Caucasian*. The distortion function $\delta$ has the following behavior. Jumps of more than one category in age and prior counts are heavily discouraged by a high distortion penalty ($10^4$) for such transformations. We impose the same penalty on increases in recidivism (change of $Y$ from 0 to 1). Both these choices are made in the interest of individual fairness. Furthermore, for every jump to an adjacent category for age and prior counts, a penalty of 1 is assessed, and a similar jump in charge degree incurs a penalty of 2. Reduction in recidivism (1 to 0) has a penalty of 2. The total distortion for each individual is the sum of squares of distortions for each attribute of $X$.

*Distortion function for Adult:* We use three conditional probability constraints of the form in (5). In constraint $i$, the distortion function returns 1 in case ($i$) and 0 otherwise: (1) if income is decreased, age is not changed and education is increased by at most 1 year, (2) if age is changed by a decade and education is increased by at most 1 year regardless of the change of income, (3) if age is changed by more than a decade or education is lowered by any amount or increased by more than 1 year. The corresponding probability bounds $c_{d,x,y}$ are 0.1, 0.05, 0 (no dependence on $d, x, y$). As a consequence, and in the same broad spirit as for COMPAS, decreases in income, small changes in age, and small increases in education (events (1), (2)) are permitted with small probabilities, while larger changes in age and education (event (3)) are not allowed at all.

Once the optimized randomized mapping $p_{\hat{X},\hat{Y}|D,X,Y}$ is determined, we apply it to the training set to obtain a new perturbed training set, which is then used to fit two classifiers: logistic regression (LR) and random forest (RF). For the test set, we first compute the test-time mapping $p_{\hat{X}|D,X}$ in (9) using $p_{\hat{X},\hat{Y}|D,X,Y}$ and $p_{D,X,Y}$ estimated from the training set. We then independently randomize each test sample $(d_i, x_i)$ using $p_{\hat{X}|D,X}$, preserving the protected variable $D$, i.e. $(d_i, x_i) \xrightarrow{p_{\hat{X}|D,X}} (d_i, \hat{x}_i)$. Each trained classifier $f$ is applied to the transformed test samples, obtaining an estimate $\widetilde{y}_i = f(d_i, \hat{x}_i)$ which is evaluated against $y_i$. These estimates induce an empirical posterior distribution given by $p_{\widetilde{Y}|D}(1|d) = \frac{1}{n_d}\sum_{\{\hat{x}_i, d_i\}: d_i = d} f(d_i, \hat{x}_i)$, where $n_d$ is the number of samples with $d_i = d$.

For the two baselines, the above procedure is repeated without data transformation except for dropping $D$ throughout for the second baseline ($D$ is still used to compute the discrimination of the resulting classifier). Due to the lack of available code, we implemented LFR ourselves in Python and solved the associated optimization problem using the SciPy package. The parameters for LFR were set as recommended in Zemel et al. [2013]: $A_z = 50$ (group fairness), $A_x = 0.01$ (individual fairness), and $A_y = 1$ (prediction accuracy). The results did not significantly change within a reasonable variation of these three parameters.

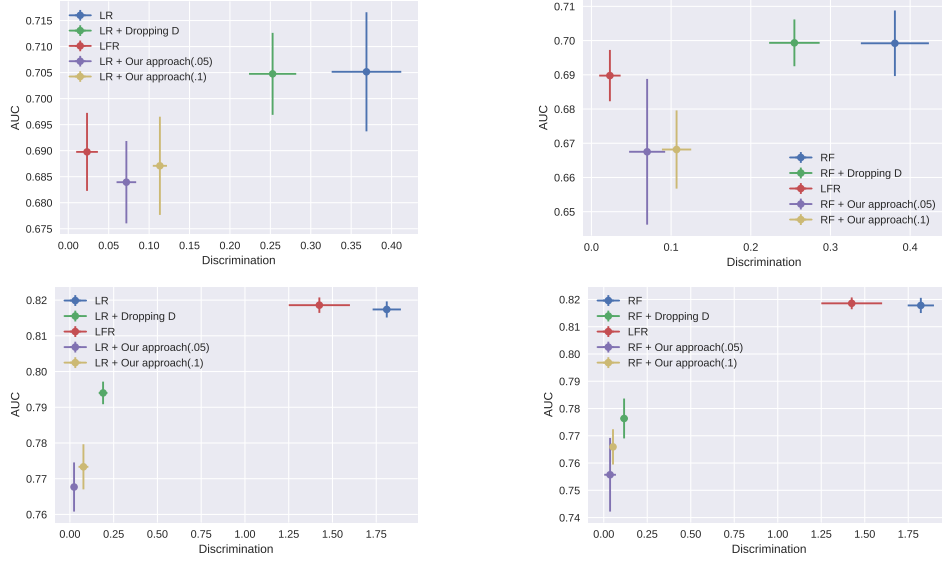

Figure 2: Discrimination-AUC plots for two different classifiers. Top row is for COMPAS dataset, and bottom row for UCI Adult dataset. First column is logistic regression (LR), and second column is random forests (RF).

**Results.** We report the trade-off between two metrics: (i) the empirical discrimination of the classifier on the test set, given by $\max_{d,d' \in \mathcal{D}} J(p_{\widetilde{Y}|D}(1|d), p_{\widetilde{Y}|D}(1|d'))$, and (ii) the empirical accuracy, measured by the Area under ROC (AUC) of $\widetilde{y}_i = f(d_i, \hat{x}_i)$ compared to $y_i$, using 5-fold cross validation. Fig. 2 presents the operating points achieved by each procedure in the discrimination-accuracy space as measured by these metrics. For the COMPAS dataset, there is significant discrimination in the original dataset, which is reflected by both LR and RF when the data is not transformed. Dropping the $D$ variable reduces discrimination with a negligible impact on classification. However discrimination is far from removed since the features $X$ are correlated with $D$, i.e. there is indirect discrimination. LFR with the recommended parameters is successful in further reducing discrimination while still achieving high prediction performance for the task.

Our proposed optimized pre-processing approach successfully decreases the empirical discrimination close to the target $\epsilon$ values (x-axis). Deviations are expected due to the approximation of $\hat{Y}$, the output of the transformation, by $\widetilde{Y}$, the output of each classifier, and also due to the randomized nature of the method. The decreased discrimination comes at an accuracy cost, which is greater in this case than for LFR. A possible explanation is that LFR is free to search across different representations whereas our method is restricted by the chosen distortion metric and having to preserve the domain of the original variables. For example, for COMPAS we heavily penalize increases in recidivism from 0 to 1 as well as large changes in prior counts and age. When combined with the other constraints in the optimization, this may alter the joint distribution after perturbation and by extension the classifier output. Increased accuracy could be obtained by relaxing the distortion constraint, as long as this is acceptable to the practitioner. We highlight again that our distortion metric was not chosen to explicitly optimize performance on this task, and should be guided by the practitioner. Nevertheless, we do successfully obtain a controlled reduction of discrimination while avoiding unwanted deviations in the randomized mapping.

For the Adult dataset, dropping the protected variable does significantly reduce discrimination, in contrast with COMPAS. Our method further reduces discrimination towards the target $\epsilon$ values. The loss of prediction performance is again due to satisfying the distortion and discrimination constraints. On the other hand, LFR with the recommended parameters provides only a small reduction in discrimination. We note that this does not contradict the results in Zemel et al. [2013], since here we have adopted a multiplicative discrimination metric (3) whereas Zemel et al. [2013] used an additive metric. Moreover, we reduced the Adult dataset to 31 binary features which is different from Zemel et al. [2013] where they additionally considered the test dataset for Adult (12661 instances) also and created 103 binary features. By varying the LFR parameters, it is possible to attain low empirical discrimination but with a large loss in prediction performance (below the plotted range). Thus, we do not claim that our method outperforms LFR since different operating points can be achieved by

adjusting parameters in either approach. In our approach however, individual fairness is explicitly maintained through the design of the distortion metric and discrimination is controlled directly by a single parameter $\epsilon$, whereas the relationship is less clear with LFR.

# 5 Conclusions

We proposed a flexible, data-driven optimization framework for probabilistically transforming data in order to reduce algorithmic discrimination, and applied it to two datasets. When used to train standard classifiers, the transformed dataset led to a fairer classification when compared to the original dataset. The reduction in discrimination comes at an accuracy penalty due to the restrictions imposed on the randomized mapping. Moreover, our method is competitive with others in the literature, with the added benefit of enabling an explicit control of individual fairness and the possibility of multivariate, non-binary protected variables. The flexibility of the approach allows numerous extensions using different measures and constraints for utility preservation, discrimination, and individual distortion control. Investigating such extensions, developing theoretical characterizations based on the proposed framework, and quantifying the impact of the transformations on additional supervised learning tasks will be pursued in future work.

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
