[Supplementary Material]

# Supplementary Material: Additional Remarks, Experiments and Proofs
# Optimized Pre-Processing for Discrimination Prevention

**Flavio P. Calmon**
Harvard University
flavio@seas.harvard.edu

**Dennis Wei**
IBM Research AI
dwei@us.ibm.com

**Bhanukiran Vinzamuri**
IBM Research AI
bhanu.vinzamuri@ibm.com

**Karthikeyan Natesan Ramamurthy**
IBM Research AI
knatesa@us.ibm.com

**Kush R. Varshney**
IBM Research AI
krvarshn@us.ibm.com

## 1 Additional Remarks

In this supplementary material, we present additional remarks, proofs and experiments related to our main submission. When referring to equations, lemmas and propositions in the supplementary material, we use the prefix "SM". Equations and references from the main paper are referred to without prefix.

### 1.1 On Suppressing the Discriminatory Variable at Apply Time

In many applications the protected variable cannot be revealed to the classification algorithm. In this case, the train-time discrimination guarantees are preserved at apply time if the Markov relationship $D \to \hat{X} \to \hat{Y}$ (i.e. $p_{\hat{Y}|\hat{X},D} = p_{\hat{Y}|\hat{X}}$) holds since, in this case,

$$p_{\widetilde{Y}|D}(\widetilde{y}|d) \approx \sum_{\hat{x}} p_{\hat{Y}|\hat{X}}(\widetilde{y}|\hat{x}) p_{\hat{X}|D}(\hat{x}|d) = p_{\hat{Y}|D}(\widetilde{y}|d). \tag{SM.1}$$

Thus, given that the distribution $p_{D,X,Y}$ is known, the guarantees provided during training still hold when applied to fresh samples if the additional constraint $p_{\hat{X},\hat{Y}|D,X,Y} = p_{\hat{Y}|\hat{X}} p_{\hat{X}|D,X,Y}$ is satisfied. We refer to (6) with the additional constraint $p_{\hat{X},\hat{Y}|D,X,Y} = p_{\hat{Y}|\hat{X}} p_{\hat{X}|D,X,Y}$ as the *suppressed optimization formulation* (SOF). Alas, since the added constraint is non-convex, the SOF is not a convex program, despite being convex in $p_{\hat{X}|D,X,Y}$ for a fixed $p_{\hat{Y}|\hat{X}}$ and vice-versa (i.e. it is biconvex). We propose next two strategies for addressing this problem.

1. The first approach is to restrict $p_{\hat{Y}|\hat{X}} = p_{Y|X}$ and solve (6) for $p_{\hat{X}|D,X,Y}$. If $\Delta(\cdot, \cdot)$ is an $f$-divergence, then

$$
\begin{aligned}
\Delta\left(p_{X,Y}, p_{\hat{X},\hat{Y}}\right) &= D_f\left(p_{X,Y} \| p_{\hat{X},\hat{Y}}\right) \\
&= \sum_{x,y} p_{\hat{X},\hat{Y}}(x,y) f\left(\frac{p_{X,Y}(x,y)}{p_{\hat{X},\hat{Y}}(x,y)}\right) \\
&\geq \sum_{x} p_{\hat{X}}(x) f\left(\sum_{y} p_{\hat{Y}|\hat{X}}(x|y) \frac{p_{X,Y}(x,y)}{p_{\hat{X},\hat{Y}}(x,y)}\right) \\
&= D_f\left(p_X \| p_{\hat{X}}\right),
\end{aligned}
$$

where the inequality follows from convexity of $f$. Since the last quantity is achieved by setting $p_{\hat{Y}|\hat{X}} = p_{Y|X}$, this choice is optimal in terms of the objective function. It may, however, render the constraints in (6) infeasible. Assuming feasibility is maintained, this approach has the added benefit that a classifier $f_\theta(x) \approx p_{Y|X}(\cdot|x)$ can be trained using the original (non-perturbed) data, and maintained for classification during apply time.

2. Alternatively, a solution can be found through alternating minimization: fix $p_{\hat{Y}|\hat{X}}$ and solve the SOF for $p_{\hat{X}|D,X,Y}$, and then fix $p_{\hat{X}|D,X,Y}$ as the optimal solution and solve the SOF for $p_{\hat{Y}|\hat{X}}$. The resulting sequence of values of the objective function is non-increasing, but may converge to a local minima.

## 1.2 A Note on Estimation and Discrimination

There is a close relationship between estimation and discrimination. If the protected variable $D$ can be reliably estimated from the outcome variable $Y$, then it is reasonable to expect that the discrimination control constraint (1) does not hold for small values of $\epsilon_{y,d}$. We make this intuition precise in the case when $J$ is given in (3) next.

More specifically, we prove that if the advantage of estimating $D$ from $Y$ over a random guess is large, then there must exist a value of $d$ and $y$ such that $J(p_{Y|D}(y|d), p_{Y_T}(y))$ is also large. Thus, standard estimation methods can be used to detect the presence of discrimination: if an estimation algorithm can estimate $D$ from $Y$, then discrimination may be present. Alternatively, if discrimination control is successful, then no estimator can significantly improve upon a random guess when estimating $D$ from $Y$.

We denote the highest probability of correctly guessing $D$ from an observation of $Y$ by $P_c(D|Y)$, where

$$P_c(D|Y) \triangleq \max_{D \to Y \to \hat{D}} \Pr\left(D = \hat{D}\right), \tag{SM.2}$$

and the maximum is taken across all estimators $p_{\hat{D}|Y}$ that satisfy the Markov condition $D \to Y \to \hat{D}$. For $D$ and $Y$ defined over finite supports, this is achieved by the maximum *a posteriori* (MAP) estimator and, consequently,

$$P_c(D|Y) = \sum_{y \in \mathcal{Y}} p_Y(y) \max_{d \in \mathcal{D}} p_{D|Y}(d|y). \tag{SM.3}$$

Let $p_D^*$ be the most likely outcome of $D$, i.e. $p_D^* \triangleq \max_{d \in \mathcal{D}} p_D(d)$. The (multiplicative) advantage over a random guess is given by

$$\mathsf{Adv}(D|Y) \triangleq \frac{P_c(D|Y)}{p_D^*}. \tag{SM.4}$$

The next proposition connects discrimination and estimation. Simply put, it shows that if a protected variable $D$ can be reliably estimated from the decision variable $Y$, then $Y$ can discriminate in terms of $D$. The proposition is given in terms of discrimination measured as (1), but a similar result holds for (2).

**Proposition SM.1.** *For $D$ and $Y$ defined over finite support sets, if $\mathsf{Adv}(D|Y) > 1 + \epsilon$, then for any $p_{Y_T}$, there exists $y \in \mathcal{Y}$ and $d \in \mathcal{D}$ such that $\left|\frac{p_{Y|D}(y|d)}{p_{Y_T}(y)} - 1\right| > \epsilon$.*

*Proof.* We show the contrapositive of the statement of the proposition. Assume that

$$\left|\frac{p_{Y|D}(y|d)}{p_{Y_T}(y)} - 1\right| \le \epsilon \ \forall y \in \mathcal{Y}, d \in \mathcal{D}. \tag{SM.5}$$

Then

$$P_c(D|Y) = \sum_{y \in \mathcal{Y}} \max_{d \in \mathcal{D}} p_{D|Y}(d|y) p_Y(y)$$

$$= \sum_{y \in \mathcal{Y}} \max_{d \in \mathcal{D}} p_{Y|D}(y|d) p_D(d)$$

$$\leq \sum_{y \in \mathcal{Y}} \max_{d \in \mathcal{D}} (1 + \epsilon) p_{Y_T}(y) p_D(d)$$

$$= (1 + \epsilon) \max_{d \in \mathcal{D}} p_D(d),$$

where the inequality follows by noting that (SM.5) implies $p_{Y|D}(y|d) \leq (1+\epsilon)p_{Y_T}(y)$ for all $y \in \mathcal{Y}$, $d \in \mathcal{D}$. Rearranging the terms of the last equality, we arrive at

$$\frac{P_c(D|Y)}{\max_{d \in \mathcal{D}} p_D(d)} \leq 1 + \epsilon,$$

and the result follows by observing that the left-hand side is the definition of $\mathsf{Adv}(D|Y)$. □

## 2 Proofs

### 2.1 Proof of Proposition 1

We restate the proposition and provide its proof below.

**Proposition 1.** *Problem* (6) *is a (quasi)convex optimization if $\Delta(\cdot, \cdot)$ is (quasi)convex and $J(\cdot, \cdot)$ is quasiconvex in their respective first arguments (with the second arguments fixed). If discrimination constraint* (2) *is used in place of* (1)*, then the condition on $J$ is that it be jointly quasiconvex in both arguments.*

*Proof.* Considering first the objective function, the distribution $p_{X,Y}$ is a given quantity while

$$p_{\hat{X},\hat{Y}}(\hat{x}, \hat{y}) = \sum_{d,x,y} p_{D,X,Y}(d, x, y) p_{\hat{X},\hat{Y}|D,X,Y}(\hat{x}, \hat{y}|d, x, y)$$

is seen to be a linear function of the mapping $p_{\hat{X},\hat{Y}|D,X,Y}$, i.e. the optimization variable. Hence if the statistical dissimilarity $\Delta(\cdot, \cdot)$ is convex in its first argument with the second fixed, then $\Delta(p_{\hat{X},\hat{Y}}, p_{X,Y})$ is a convex function of $p_{\hat{X},\hat{Y}|D,X,Y}$ by the affine composition property [Boyd and Vandenberghe, 2004]. This condition is satisfied for example by all $f$-divergences [Csiszár and Shields, 2004], which are jointly convex in both arguments, and by all Bregman divergences [Banerjee et al., 2005]. If instead $\Delta(\cdot, \cdot)$ is only quasiconvex in its first argument, a similar composition property implies that $\Delta(p_{\hat{X},\hat{Y}}, p_{X,Y})$ is a quasiconvex function of $p_{\hat{X},\hat{Y}|D,X,Y}$ [Boyd and Vandenberghe, 2004].

For discrimination constraint (1), the target distribution $p_{Y_T}$ is assumed to be given. The conditional distribution $p_{\hat{Y}|D}$ can be related to $p_{\hat{X},\hat{Y}|D,X,Y}$ as follows:

$$p_{\hat{Y}|D}(\hat{y}|d) = \sum_{\hat{x}} \sum_{x,y} p_{X,Y|D}(x, y|d) p_{\hat{X},\hat{Y}|D,X,Y}(\hat{x}, \hat{y}|d, x, y).$$

Since $p_{X,Y|D}$ is given, $p_{\hat{Y}|D}$ is a linear function of $p_{\hat{X},\hat{Y}|D,X,Y}$. Hence by the same composition property as above, (1) is a convex constraint, i.e. specifies a convex set, if the distance function $J(\cdot, \cdot)$ is quasiconvex in its first argument.

If constraint (2) is used instead of (1), then both arguments of $J$ are linear functions of $p_{\hat{X},\hat{Y}|D,X,Y}$. Hence (2) is convex if $J$ is jointly quasiconvex in both arguments.

Lastly, the distortion constraint (4) can be expanded explicitly in terms of $p_{\hat{X},\hat{Y}|D,X,Y}$ to yield

$$\sum_{\hat{x},\hat{y}} p_{\hat{X},\hat{Y}|D,X,Y}(\hat{x}, \hat{y}|d, x, y) \delta((x, y), (\hat{x}, \hat{y})) \leq c_{d,x,y}.$$

Thus (4) is a linear constraint in $p_{\hat{X},\hat{Y}|D,X,Y}$ regardless of the choice of distortion metric $\delta$. □

### 2.2 Proof of Proposition 2

We restate the proposition next.

**Proposition 2.** *Let $p_{D,X,Y}$ be the empirical distribution obtained from $n$ i.i.d. samples that is used to determine the mapping $p_{\hat{Y},\hat{X}|X,Y,D}$, and $q_{D,X,Y}$ be the true distribution of the data. In addition, denote by $q_{D,\hat{X},\hat{Y}}$ the joint distribution after applying $p_{\hat{Y},\hat{X}|X,Y,D}$ to samples from $q_{D,X,Y}$. If for all $y \in \mathcal{Y}$, $d \in \mathcal{D}$ we have $p_{Y,D}(y,d) > 0$, $J\left(p_{\hat{Y}|D}(y|d), p_{Y_T}(y)\right) \leq \epsilon$, where $J$ is given in* (3), *and*

$$\Delta\left(p_{X,Y}, p_{\hat{X},\hat{Y}}\right) = \sum_{x,y} \left| p_{X,Y}(x,y) - p_{\hat{X},\hat{Y}}(x,y) \right| \leq \mu,$$

*then with probability $1 - \beta$,*

$$\max\left\{ J\left(q_{\hat{Y}|D}(y|d), p_{Y_T}(y)\right) - \epsilon, \Delta\left(q_{X,Y}, q_{\hat{X},\hat{Y}}\right) - \mu \right\} \lesssim \sqrt{\frac{m}{n}\log\left(1 + \frac{n}{m}\right) - \frac{\log\beta}{n}}.$$
$$\text{(SM.6)}$$

As discussed in the main text, observe that hidden within the big-O terms in Proposition 2 are constants that depend on the probability of the least likely symbol and the alphabet size. Moreover, the upper bounds become loose if $p_{Y,D}(y,d)$ can be made arbitrarily small. Thus, it is necessary to assume that $p_{Y,D}(y,d)$ is fixed and bounded away from zero. If the dimensionality of the support sets of $D, X$ and $Y$ is large, and the number of samples $n$ is limited, then a dimensionality reduction step (e.g. clustering) may be necessary in order to assure that discrimination control and utility are adequately preserved at test time. Proposition 2 and its proof can be used to provide an explicit estimate of the required reduction. It is also clear from the proof that an equivalent upper-bound on the convergence rate can be obtained when the discrimination constraint is of the form (2), i.e. for all $y \in \mathcal{Y}$, $d \in \mathcal{D}$ we have $J\left(p_{\hat{Y}|D}(y|d_1), p_{Y_T}(y|d_2)\right) \leq \epsilon$.

Finally, we also note that if there are insufficient samples to reliably estimate $q_{D,X,Y}(d,x,y)$ for certain values $(d,x,y) \in \mathcal{D} \times \mathcal{X} \times \mathcal{Y}$, then, for those groups $(d,x)$, it is statistically challenging to verify discrimination and thus control may not be meaningful.

The proposition is a consequence of the following elementary lemma.

**Lemma SM.1.** *Let $p(x)$, $q(x)$ and $r(x)$ be three fixed probability mass functions with the same discrete and finite support set $\mathcal{X}$, $c_1 \triangleq \min_{x \in \mathcal{X}} \frac{p(x)(1-p(x))}{3(1+p(x))^2} > 0$ and $p_m \triangleq \min_x p(x) > 0$. Then if*

$$D_{\mathsf{KL}}(p\|q) \leq \tau \leq c_1 \qquad \text{(SM.7)}$$

*and for all $x \in \mathcal{X}$ and*

$$\gamma_1 \leq \frac{p(x)}{r(x)} \leq \gamma_2, \qquad \text{(SM.8)}$$

*then for all $x \in \mathcal{X}$ and $g(\tau, p_m) \triangleq \sqrt{\frac{3\tau}{p_m}}$*

$$\gamma_1 \exp\left(-g(\tau, p_m)\right) \leq \frac{q(x)}{r(x)} \leq \gamma_2 \exp\left(g(\tau, p_m)\right). \qquad \text{(SM.9)}$$

*Proof.* We assume $\tau > 0$, otherwise $p(x) = q(x)$ $\forall x \in \mathcal{X}$ and we are done. From (SM.7) and the Data Processing Inequality for KL-divergence, for any $x \in \mathcal{X}$

$$p(x)\log\frac{p(x)}{q(x)} + (1-p(x))\log\frac{1-p(x)}{1-q(x)} \leq \tau. \qquad \text{(SM.10)}$$

Let $x$ be fixed, and, in order to simplify notation, denote $c \triangleq p(x)$. Assuming, without loss of generality,

$$q(x) = c\exp\left(-\frac{\alpha\tau}{c}\right),$$

then (SM.10) implies

$$f(\alpha) \triangleq \alpha - \frac{1-c}{\tau}\log\left(\frac{1 - c\exp\left(-\frac{\alpha\tau}{c}\right)}{1-c}\right) \leq 1. \qquad \text{(SM.11)}$$

The Taylor series of $f(\alpha)$ around 0 has the form

$$f(\alpha) = \sum_{n=2}^{\infty} \frac{(-1)^n}{n!} \left( \frac{\tau}{(1-c)c} \right)^{n-1} A_{n-1}(c)\alpha^n, \tag{SM.12}$$

where $A_n(c)$ is the Eulerian polynomial, which is positive for $c > 0$ and satisfies $A_1(c) = 1$ and $A_2(c) = (1+c)$. First, assume $\alpha \leq 0$. Then $f(\alpha)$ can be lower-bounded by the first term in its Taylor series expansion since all the terms in the series are non-negative. From (SM.11),

$$\frac{\tau\alpha^2}{2(1-c)c} \leq f(\alpha) \leq 1. \tag{SM.13}$$

Consequently,

$$\alpha \geq -\sqrt{\frac{2(1-c)c}{\tau}}. \tag{SM.14}$$

Now assume $\alpha \geq 0$. Then the Taylor series (SM.12) becomes an alternating series, and $f(\alpha)$ can be lower-bounded by its first two terms

$$\frac{\tau\alpha^2}{2(1-c)c} - \frac{(1+c)\tau^2\alpha^3}{6(1-c)^2c^2} \leq f(\alpha) \leq 1. \tag{SM.15}$$

The term in the l.h.s. of the first inequality satisfies

$$\frac{\tau\alpha^2}{3(1-c)c} \leq \frac{\tau\alpha^2}{2(1-c)c} - \frac{(1+c)\tau^2\alpha^3}{6(1-c)^2c^2} \tag{SM.16}$$

as long as $\alpha \leq \frac{c(1-c)}{(1+c)\tau}$. Since the lhs is larger than 1 when $\alpha > \sqrt{\frac{3(1-c)c}{\tau}}$, then it is a valid lower-bound for $f(\alpha)$ in the entire interval where $f(\alpha) \leq 1$ and $\alpha \geq 0$ as long as

$$\sqrt{\frac{3(1-c)c}{\tau}} \leq \frac{c(1-c)}{(1+c)\tau} \Leftrightarrow \tau \leq \frac{c(1-c)}{3(1+c)^2}, \tag{SM.17}$$

which holds by assumption in the Lemma. Thus,

$$\alpha \leq \sqrt{\frac{3(1-c)c}{\tau}}, \tag{SM.18}$$

and combining the previous equation with (SM.14)

$$-\sqrt{\frac{2(1-c)c}{\tau}} \leq \alpha \leq \sqrt{\frac{3(1-c)c}{\tau}} \tag{SM.19}$$

Finally, since $\frac{q(x)}{p(x)} = \exp(-\alpha\tau/p(x))$, from the previous inequalities

$$\exp\left( -\sqrt{\frac{3(1-p(x))\tau}{p(x)}} \right) \leq \frac{q(x)}{p(x)}$$
$$\leq \exp\left( \sqrt{\frac{2(1-p(x))\tau}{p(x)}} \right), \tag{SM.20}$$

and the result follows by further lower bounding the lhs by $\gamma_1 r(x) \leq p(x)$ and upper bounding the rhs by $p(x) \geq \gamma_2 r(x)$ $\qquad\square$

The previous Lemma allows us to derive the result presented in Proposition 2.

*Proof of Proposition 2.* Let $m \triangleq |\mathcal{X}||\mathcal{Y}||\mathcal{D}|$. The distribution $p_{D,X,Y}$ is the type [Cover and Thomas, 2006][Chap. 11] of $n$ observations of $q_{D,X,Y}$. Then[1], from [Csiszár and Shields, 2004][Corollary 2.1], for $\tau > 0$

$$\Pr\left( D_{\mathsf{KL}}(p_{D,X,Y} \| q_{D,X,Y}) \geq \tau \right) \leq \binom{n+m-1}{m-1} e^{-n\tau}$$

$$\leq \left(\frac{e(n+m)}{m}\right)^m e^{-n\tau}.$$

From the Data Processing Inequality for KL-divergence, if $D_{\mathsf{KL}}(p_{D,\hat{Y}}\|q_{D,\hat{Y}}) \leq D_{\mathsf{KL}}(p_{D,X,Y}\|q_{D,X,Y})$, and, consequently,

$$\Pr\left(D_{\mathsf{KL}}(p_{D,\hat{Y}}\|q_{D,\hat{Y}}) \leq \tau\right) \geq \Pr\left(D_{\mathsf{KL}}(p_{D,X,Y}\|q_{D,X,Y}) \leq \tau\right)$$

$$\geq 1 - \left(\frac{e(n+m)}{m}\right)^m e^{-n\tau}.$$

If $D_{\mathsf{KL}}(p_{D,\hat{Y}}\|q_{D,\hat{Y}}) \leq \tau$, then since $0 \leq D_{\mathsf{KL}}(p_D\|q_D)$, we have

$$D_{\mathsf{KL}}(p_{\hat{Y}|D}(\cdot|d)\|q_{\hat{Y}}(\cdot|d)) \leq \frac{\tau}{p_D(d)} \ \forall d \in \mathcal{D}.$$

Choosing

$$\tau = \frac{1}{n}\log\left(\frac{1}{\beta}\left(\frac{e(n+m)}{m}\right)^m\right), \tag{SM.21}$$

then, with probability $1 - \beta$, for all $d \in D$

$$D_{\mathsf{KL}}(p_{\hat{Y}|D}(\cdot|d)\|q_{\hat{Y}}(\cdot|d))$$

$$\leq \frac{1}{np_D(d)}\log\left(\frac{1}{\beta}\left(\frac{e(n+m)}{m}\right)^m\right).$$

Assuming that $m$, and $c_m \triangleq \min_{y\in\mathcal{Y},d\in\mathcal{D}} p_{D,\hat{Y}}(d,y) > 0$ constant, from the proof of Lemma SM.1 and, more specifically, inequalities (SM.20), as long as $\tau \leq \min_{d,y} \frac{p_{\hat{Y},D}(y,d)(1-p_{\hat{Y}|D}(y|d))}{3(1+p_{\hat{Y}|D}(y|d))^2}$,

$$(1-\epsilon)\exp\left(-h(n,\beta)\right) \leq \frac{q_{\hat{Y}|D}(y|d)}{p_{Y_T}(y)} \tag{SM.22}$$

$$\leq (1+\epsilon)\exp\left(h(n,\beta)\right), \tag{SM.23}$$

where

$$h(n,\beta) \triangleq \sqrt{\frac{3}{nc_m}\log\left(\frac{1}{\beta}\left(\frac{e(n+m)}{m}\right)^m\right)}. \tag{SM.24}$$

Observe that

$$h(n,\beta) = \Theta\left(\sqrt{\frac{1}{n}\log\frac{n}{\beta}}\right).$$

Since for $x$ sufficiently small $e^x \approx 1 + x$, we have

$$\frac{\left|q_{\hat{Y}|D}(y|d) - p_{Y_T}(y)\right|}{p_{Y_T}(y)} \leq \epsilon + \Theta\left(\sqrt{\frac{1}{n}\log\frac{n}{\beta}}\right), \tag{SM.25}$$

proving the first claim.

For the second claim, we start by applying the triangle inequality:

$$\Delta\left(q_{X,Y},q_{\hat{X},\hat{Y}}\right) \leq \Delta\left(p_{X,Y},p_{\hat{X},\hat{Y}}\right) + \Delta\left(q_{X,Y},p_{X,Y}\right)$$

$$+ \Delta\left(q_{\hat{X},\hat{Y}},p_{\hat{X},\hat{Y}}\right)$$

$$\leq \mu + \Delta\left(q_{X,Y},p_{X,Y}\right)$$

$$+ \Delta\left(q_{\hat{X},\hat{Y}},p_{\hat{X},\hat{Y}}\right). \tag{SM.26}$$

Now assume $D_{\mathsf{KL}}(p_{D,X,Y}\|q_{D,X,Y}) \leq \tau$. Then the Data Processing Inequality for KL-divergence yields $D_{\mathsf{KL}}(p_{X,Y}\|q_{X,Y}) \leq \tau$ and $D_{\mathsf{KL}}(p_{\hat{X},\hat{Y}}\|q_{\hat{X},\hat{Y}}) \leq \tau$. In addition, from Pinsker's inequality,

$$\Delta\left(q_{X,Y},p_{X,Y}\right) \leq 2\sqrt{2D_{\mathsf{KL}}(p_{X,Y}\|q_{X,Y})}$$

$$\leq 2\sqrt{2\tau},$$

and, analogously, $\Delta\left(q_{\hat{X},\hat{Y}}, p_{\hat{X},\hat{Y}}\right) \leq 2\sqrt{2\tau}$. Thus (SM.26) becomes

$$\Delta\left(q_{X,Y}, q_{\hat{X},\hat{Y}}\right) \leq \mu + 4\sqrt{2\tau}. \tag{SM.27}$$

Selecting $\tau$ as in (SM.21), then, with probability $1 - \beta$,

$$\Delta\left(q_{X,Y}, q_{\hat{X},\hat{Y}}\right) \leq \mu + 4\sqrt{\frac{2}{n}\log\left(\frac{1}{\beta}\left(\frac{e(n+m)}{m}\right)^m\right)}, \tag{SM.28}$$

and the result follows. $\qquad\square$

## 3 ProPublica's COMPAS Dataset

Recidivism refers to a person's relapse into criminal behavior. It has been found that about two-thirds of prisoners in the US are re-arrested after release [Durose et al., 2014]. It is important therefore to understand the recidivistic tendencies of incarcerated individuals who are considered for release at several points in the criminal justice system (bail hearings, parole, etc.). Automated risk scoring mechanisms have been developed for this purpose and are currently used in courtrooms in the US, in particular the proprietary COMPAS tool by Northpointe [Northpointe Inc.].

Recently, ProPublica published an article that investigates racial bias in the COMPAS algorithm [ProPublica, 2016], releasing an accompanying dataset that includes COMPAS risk scores, recidivism records, and other relevant attributes [ProPublica, 2017]. A basic finding is that the COMPAS algorithm tends to assign higher scores to African-American individuals, a reflection of the *a priori* higher prevalence of recidivism in this group. The article goes on to demonstrate unequal false positive and false negative rates between African-Americans and Caucasian-Americans, which has since been shown by Chouldechova [2016] to be a necessary consequence of the calibration of the model and the difference in a priori prevalence. In follow-up work, Chouldechova [2016] and others analyzed the recidivism and COMPAS scores and showed that, within a given COMPAS score range, the true recidivism rate is approximately independent of race, i.e. the score is well-calibrated. Chouldechova [2016] further showed that a consequence of this calibration is that certain demographic groups may experience higher false positive or false negative rates because their group has higher or lower *a priori* prevalence of recidivism.

In this work, our interest is not in the debate surrounding the COMPAS algorithm but rather in the underlying recidivism data [ProPublica, 2017]. Using the proposed data transformation approach, we demonstrate the technical feasibility of mitigating the disparate impact of recidivism records on different demographic groups while also preserving utility and individual fairness. (We make no comment on the associated societal considerations.) Considering only shown African-American and Caucasian individuals, Chouldechova [2016] show that the true recidivism rates for a group of individuals with the same COMPAS rank is approximately independent of the race, i.e., $p(\text{recidivism}|\text{compas rank}, \text{race})$ is close for both the races. However, members of certain races can still be penalized more heavily because their group has a higher recidivism prevalence, $p(\text{recidivism}|\text{race})$. We set out to mitigate this disparate impact, while preserving utility and individual fairness using our proposed data transformation approach.

As discussed in the main text, we select severity of charge, number of prior crimes, and age category to be the decision variables $(X)$. The outcome variable $(Y)$ is a binary indicator of whether the individual recidivated (re-offended), and race and gender are set to be the protected variables $(D)$[2]. The encoding of the decision and discrimination variables is described in Table 1. The dataset was processed to contain around 5k records.

## 4 Additional Applications to Datasets

We present next additional results derived by applying the proposed optimization formulation to the COMPAS and the UCI Adult dataset. These numerical results aim to demonstrate the versatility

Table 1: ProPublica dataset features.

| Feature | Values | Comments |
|---|---|---|
| Recidivism (binary) | $\{0, 1\}$ | 1 if re-offended, 0 otherwise |
| Gender | {Male, Female} | |
| Race | {Caucasian, African-American} | Races with small samples removed |
| Age category | $\{< 25, 25 - 45, > 45\}$ | years of age |
| Charge degree | {Felony, Misdemeanor} | For the current arrest |
| Prior counts | $\{0, 1 - 3, > 3\}$ | Number of prior crimes |

Figure 1: Objective vs. discrimination parameter $\epsilon$ for distortion constraint $c = 0.25$.

of our formulation in terms of utility metrics (here we use KL-divergence in addition to the total variation metric used in the main text), and illustrate the trade-off between discrimination control and utility. In addition, we also discuss the optimized probabilistic mappings produced by our formulation for both datasets. These mappings capture different discrimination patterns and societal biases that exist within each dataset.

## 4.1 A Closer Look at Randomized Mappings for ProPublica's COMPAS Recidivism Data

Using the proposed data transformation approach, we demonstrate the technical feasibility of mitigating the disparate impact of recividism records on different demographic groups while also preserving utility and individual fairness. (We make no comment on the associated societal considerations.)

**Specific Form of Optimization.** For the results in this section, we specialize our general formulation in (6) by setting the utility measure $\Delta(p_{X,Y}, p_{\hat{X},\hat{Y}})$ to be the KL divergence $D_{\mathsf{KL}}(p_{X,Y}\|p_{\hat{X},\hat{Y}})$. For discrimination control, we use (2), with $J$ given in (3), while fixing $\epsilon_{y,d_1,d_2} = \epsilon$. For the sake of simplicity, we use the expected distortion constraint in (4) with $c_{d,x,y} = c$ uniformly. For the next results, the distortion function $\delta$ in (4) has the following form. Jumps of more than one category in age and prior counts are heavily discouraged by setting a high distortion penalty ($10^4$) for such transformations. We impose the same penalty on increases in recidivism (change of $Y$ from 0 to 1). Both these choices are made to promote individual fairness. Furthermore, for every jump to the next category for age and prior counts, a penalty of 1 is assessed, and a similar jump incurs a penalty of 2 for charge degree. Reduction in recidivism (1 to 0) has a penalty of 2. The total distortion for each individual is the sum of squares of distortions for each attribute of $X$. These distortion values were chosen for demonstration purposes to be reasonable to our judgement, and can easily be tuned according to the needs of a practitioner.

**Results.** We computed the optimal objective value (i.e., KL divergence) resulting from solving (6) for different values of the discrimination control parameter $\epsilon$, when the expected distortion constraint $c = 0.25$. Around $\epsilon = 0.2$, no feasible solution can be found that also satisfies the distortion constraint. Above $\epsilon = 0.59$, the discrimination control is loose enough to be satisfied by the original dataset with just an identity mapping ($D_{\mathsf{KL}}(p_{X,Y}\|p_{\hat{X},\hat{Y}}) = 0$). In between, the optimal value varies as a smooth function (Fig. 1).

Figure 2: Conditional mappings $p_{\hat{X},\hat{Y}|X,Y,D}$ with $\epsilon = 0.1$, and $c = 0.5$ for: (**left**) $D = $ (African-American, Male), less than 25 years ($X$), $Y = 1$, (**middle**) $D = $ (African-American, Male), less than 25 years ($X$), $Y = 0$, and (**right**) $D = $ (Caucasian, Male), less than 25 years ($X$), $Y = 1$. Original charge degree and prior counts ($X$) are shown in vertical axis, while the transformed age category, charge degree, prior counts and recidivism $(\hat{X}, \hat{Y})$ are represented along the horizontal axis. The charge degree F indicates felony and M indicates misdemeanor. Colors indicate mapping probability values. Columns included only if the sum of its values exceeds 0.05.

Figure 3: Top row: Percentage recidivism rates in the original dataset as a function of charge degree, age and prior counts for the overall population (i.e. $p_{Y|X}(1|x)$) and for different groups ($p_{Y|X,D}(1|x,d)$). Bottom row: Change in percentages due to transformation, i.e. $p_{\hat{Y}|\hat{X},D}(1|x,d) - p_{Y|X,D}(1|x,d)$, etc. Values for cohorts of charge degree, age, and prior counts with fewer than 20 samples are not shown. The discrimination and distortion constraints are set to $\epsilon = 0.1$ and $c = 0.5$ respectively.

We set $c = 0.5$ and $\epsilon = 0.1$ for the rest of the experiments. The optimal value of utility measure (KL divergence) was $0.021$. In order to evaluate if discrimination control was achieved as expected, we examine the dependence of the outcome variable on the discrimination variable before and after the transformation. Note that to have zero disparate impact, we would like the $\hat{Y}$ to be independent of $D$, but practically it will be controlled by the discrimination control parameter $\epsilon$. The corresponding marginals $p_{Y|D}$ and $p_{\hat{Y}|D}$ are illustrated in Table 2, where clearly $\hat{Y}$ is less dependent on $D$ compared to $Y$. In particular, since an increase in recidivism is heavily penalized, the net effect of the randomized transformation is to decrease the recidivism risk of males, and particularly African-American males.

The mapping $p_{\hat{X},\hat{Y}|X,Y,D}$ produced by the optimization (6) can reveal important insights on the nature of disparate impact and how to mitigate it. We illustrate this by exploring $p_{\hat{X},\hat{Y}|X,Y,D}$ for the COMPAS dataset next. Fig. 2 displays the conditional mapping restricted to certain socio-

Table 2: Dependence of the outcome variable on the discrimination variable before and after the proposed transformation. F and M indicate Female and Male, and A-A, and C indicate African-American and Caucasian.

| D | Before transformation | | After transformation | |
|---|---|---|---|---|
| (gender, race) | $p_{Y\|D}(0\|d)$ | $p_{Y\|D}(1\|d)$ | $p_{\hat{Y}\|D}(0\|d)$ | $p_{\hat{Y}\|D}(1\|d)$ |
| F, A-A | 0.607 | 0.393 | 0.607 | 0.393 |
| F, C | 0.633 | 0.367 | 0.633 | 0.367 |
| M, A-A | 0.407 | 0.593 | 0.596 | 0.404 |
| M, C | 0.570 | 0.430 | 0.596 | 0.404 |

Figure 4: Top row: High income percentages in the original dataset as a function of age and education for the overall population (i.e. $p_{Y\|X}(1\|x)$) and for different groups $p_{Y\|X,D}(1\|x,d)$). Bottom row: Change in percentages due to transformation, i.e. $p_{\hat{Y}\|\hat{X},D}(1\|x,d) - p_{Y\|X,D}(1\|x,d)$, etc. Age-education pairs with fewer than 20 samples are not shown.

demographic groups. First consider young males who are African-American (left-most plot). This group has a high recidivism rate, and hence the most prominent action of the mapping (besides identity transformation) is to change the recidivism value from 1 (recidivism) to 0 (no recidivism). The next prominent action is to change the age category from young to middle aged (25 to 45 years). This effectively reduces the average value of $\hat{Y}$ for young African-Americans, since the mapping for young males who are African-American and do not recidivate (middle plot) is essentially the identity mapping, with the exception of changing age category to middle aged. This is expected, since increasing recidivism is heavily penalized. For young Caucasian males who recidivate, the action of the proposed transformation seems to be similar to that of young African-American males who recidivate, i.e., the outcome variable is either changed to 0, or the age category is changed to middle age. However the probabilities of the transformations are lower since Caucasian males have, according to the dataset, a lower recidivism rate.

We apply this conditional mapping on the dataset (one trial) and present the results in Fig. 3. The original percentage recidivism rates are also shown in the top panel of the plot for comparison. Because of our constraint that disallows changing the outcome to 1, a demographic group's recidivism rate can (indirectly) increase only through changes to the decision variables ($X$). We note that the average percentage change in recidivism rates across all demographics is negative when the discrimination variables are marginalized out (leftmost column). The maximum decreases in recidivism rates are observed for African-American males since they have the highest value of $p_{Y\|D}(1\|d)$ (cf. Table 2). Contrast this with Caucasian females (middle column), who have virtually no change in their recidivism rates since they are a priori close to the final ones (see Table 2). Another interesting observation is that middle aged Caucasian males with 1 to 3 prior counts see an increase in percentage recidivism. This is consistent with the mapping seen in Fig. 2 (middle), and is an example of the indirect introduction of positive outcome variables in a cohort as discussed above.

## 4.2 Comparison Between the Proposed Method and LFR on the UCI Adult Data Set

The number of prototypes $K$ was set to 10 while running LFR on the Adult dataset. We followed the parameter settings in the SM of their original paper and set the values for both $A_x$ and $A_z$ at 0.01 and 50, respectively. We performed parameter tuning for $A_y$ over the set {0.1,0.5,1,5,10} optimizing for prediction accuracy.

## 4.3 A Closer Look at Randomized Mappings for the UCI Adult Data Set

We now look at the randomized mappings produced for the UCI Adult Dataset [Lichman, 2013]. Here, the features were categorized as protected variables ($D$): Race (White, Minority) and Gender (Male, Female)[3]; decision variables ($X$): Age (quantized to decades) and Education (quantized to years); and response variable ($Y$): Income (binary). While the response variable considered here is income, the dataset could be regarded as a simplified proxy for analyzing other financial outcomes such as credit approvals.

**Specific Form of Optimization.** For the next collection of results, we use $\ell_1$-distance (twice the total variation) [Pollard, 2002] to measure utility, $\Delta\left(p_{X,Y}, p_{\hat{X},\hat{Y}}\right) = \sum_{x,y}\left|p_{X,Y}(x,y) - p_{\hat{X},\hat{Y}}(x,y)\right|$. For discrimination control, we use (1), with $J$ given in (3), We also set $\epsilon_{y,d} = \epsilon$ in (1). We use the distortion function in (4), and write $x = (a,e)$ for an age-education pair and $\hat{x} = (\hat{a},\hat{e})$ for a corresponding transformed pair. The distortion function returns (i) $v_1$ if income is decreased, age is not changed and education is increased by at most 1 year, (ii) $v_2$ if age is changed by a decade and education is increased by at most 1 year regardless of the change of income, (iii) $v_3$ if age is changed by more than a decade or education is lowered by any amount or increased by more than 1 year, and (iv) 0 in all other cases. We set $(v_1, v_2, v_3) = (1, 2, 3)$ with corresponding distance thresholds for $\delta = 0$ as $(0.9, 1.9, 2.9)$ and corresponding probabilities $(c_{d,x,y})$ as $(0.1, 0.05, 0)$ in (4). As a consequence, decreases in income, small changes in age, and small increases in education (events (i), (ii)) are permitted with small probabilities, while larger changes in age and education (event (iii)) are not allowed at all. We note that the parameter settings are selected with the purpose of demonstrating our approach, and would change depending on the practitioner's requirements or guidelines.

**Results.** For the remainder of this section, we set $\epsilon = 0.15$, and the optimal value of the utility measure ($\ell_1$ distance) was 0.014. We apply the conditional mapping, generated as the optimal solution to (6), to transform the age, education, and income values of each sample in the dataset. The result of a single realization of this randomization is given in Fig. 4, where we show percentages of high income individuals as a function of age and education before and after the transformation. The original age and education ($X$) are plotted throughout Fig. 4 for ease of comparison, and that changes in individual percentages may be larger than a factor of $1 \pm \epsilon$ because discrimination is not controlled by (1) at the level of age-education cohorts. The top left panel indicates that income is higher for more educated and middle-aged people, as expected. The second column shows that high income percentages are significantly lower for females and are accordingly increased by the transformation, most strongly for educated older women and younger women with only 8 years of education, and less so for other younger women. Conversely, the percentages are decreased for males but by much smaller magnitudes. Minorities receive small percentage increases but less than for women, in part because they are a more heterogeneous group consisting of both genders.

## 4.4 Final Remarks

The differences between the original and transformed datasets revealed interesting discrimination patterns, as well as corrective adjustments for controlling discrimination while preserving utility of the data. Despite being programmatically generated, the optimized transformation satisfied properties that are sensible from a socio-demographic standpoint, reducing, for example, recidivism risk for males who are African-American in the recidivism dataset, and increasing income for well-educated females in the UCI adult dataset.

## Footnotes

[1]Other bounds on the KL-divergence between an observed type and its distribution could be used, such as [Cover and Thomas, 2006][Thm. 11.2.2], without changing the asymptotic result.

[2]In the main text we considered only race as the protected variable $D$. Here, we consider both race and gender.

[3]In the main text we only consider Gender as a protected variable. Here we consider both Gender and Race.