[Reviews · NeurIPS 2017]

Reviewer 1



This paper introduces a new framework to convert a dataset so that the converted dataset satisfies both group fairness and individual fairness. The present framework is formulated as a convex optimization problem under certain conditions. The authors prove a generalization error bounds in terms of utility and group fairness. Furthermore, they confirmed the performance of their proposed framework by experiments. First of all, the contributions of this paper are not clearly explained. The contributions should be more clearly and precisely described in the introduction. $ Details In the theoretical viewpoint, the convexity in Proposition 1 is actually obvious. The authors should discuss that the convexity of the proposed framework has a considerable impact on the tradeoff between utility, group fairness, and individual fairness. Proposition 2 is actually confusing. It ignores the most important term $m$ in its proof by introducing the big-O notion. The bounds should be $O(\sqrt{m log(1 + n/m)/n + log(1/\beta)/n})$. The first term in the square root denotes the complexity term since $m$ denotes the parameter size of the learning transformation. For example, let the non-discrimination variable form a $d$ dimensional binary vector. Then, $m \ge 2^d$, and thus it requires exponential samples with respect to $d$ to achieve the constant order bound. In the sense of the sample complexity, the present framework does not satisfy learnability. The empirical comparisons between the present method and LFR are not fair. The LFR has tunable parameters to adjust trade-off between utility, group fairness and individual fairness, and thus these parameters should be tuned in an appropriate way. Moreover, the authors should conduct experiments with varying values of \epsilon more finely. The experiments presented in the manuscript cannot support the claim that the proposed framework can control the tradeoff between utility and fairness. Furthermore, the authors should compare the present method with LFR in the sense of individual fairness. There is no evidence that the present method can ensure the individual fairness. Thus, the significance of the contribution seems to be weak in terms of theory and experiments in its current form. ---- After rebuttal discussion, I become more positive on this paper and raised the score.

Reviewer 2



This paper presents a constrained optimization framework to address (ethically/socially sensitive) discrimination in a classification setting. Such discrimination can occur in machine learning because (1) the training data was obtained from an unfair world and reflects that unfairness, (2) the algorithm trained on the data may introduce biases or unfairness even if there are no issues in the training data, or (3) the output is used in a discriminatory fashion. The present paper falls in the category of preprocessing, meaning it attempts to address unfairness coming from (1) by transforming the data to reduce or eliminate its discriminatory aspects. As notation, D represents one or more variables of a sensitive nature for which we wish to reduce discrimination, such as race or sex, X represents other predictors, and Y the outcome variable for (binary) classification. Existing literature on fairness and discrimination contains a variety of formal definitions of (un)fairness. Two such definitions are used here, (I) dependence of the predicted outcome \hat Y on D should be low, and (II) similar individuals should receive similar predicted outcomes. These two criteria form the constraints of the optimization problem posed in this paper, and the objective is to minimize a probability distance measure between the joint distribution of the transformed data and the joint distribution of the original data. The paper gives conditions under which the optimization problem is (quasi)convex, and proves a result bounding errors resulting from misspecification of the prior distribution of the data. The utility of the method is demonstrated empirically on two datasets and shown to be competitive with the most similar existing method of Zemel et al. (2013). One important limitation of this paper is the assumption that the joint distribution of the data is known. Proposition 2 is a good attempt at addressing this limitation. However, there may be subtle issues related to fairness encoded in the joint distribution which are only hinted at in the Supplementary Material in Figures 2-4. For example, what does it mean for two individuals with different values of D to be similar otherwise? Are there important aspects of similarity not captured in the available data--i.e. confounding variables? These questions are fundamental limitations of individual fairness (IF), and while I may have objections to (IF) I recognize it has been accepted in previous literature and is not a novel proposal of the present work. However, even accepting (IF) without objections, Proposition 2 appears only to address the other two aspects of the optimization problem, and does not address the effect of prior distribution misspecification on (IF). If this could be changed it would strengthen the paper. Some acknowledgement of the subtlety in defining fairness--issues concerning the relations between D and other variables in X, and the implications of those relations on Y, which are addressed fairly implicitly in (IF)--would also strengthen the exposition. See, for example, Johnson et al. "Impartial Predictive Modeling: Ensuring Fairness in Arbitrary Models," or Kusner et al. "Counterfactual Fairness." Another limitation of the present work is the large amount of calibration or tuning parameters involved. There are tolerances for each constraint, and the potentially contradictory nature of the two kinds of constraints (discrimination vs distortion) is not acknowledged except in the Supplementary Materials where Figure 1 shows some constraint tolerances may be infeasible. Addressing this in the main text and giving conditions (perhaps in the Supplementary Materials) under which the optimization problem is feasible would also strengthen the paper. Finally, the choice of the distortion metric is both a subtle issue where the fundamental goal of fairness is entirely implicit--a limitation of (IF)--and a potentially high dimensional calibration problem. The metric can be customized on a per-application basis using domain and expert knowledge, which is both a limitation and a strength. How should this be done so the resulting definition of (IF) is sensible? Are there any guidelines or default choices, or references to other works addressing this? Figure 2 is rather uninformative since neither the present method nor its competitor have been tuned. The comparison is (probably) fair, but the reader is left wondering what the best performance of either method might look like, and whether one strictly dominates the other. * Quality The paper is well supported by theoretical analysis and empirical experiments, especially when the Supplementary Materials are included. The work is fairly complete, though as mentioned above it is both a strength and a weakness that there is much tuning and other specifics of the implementation that need to be determined on a case by case basis. It could be improved by giving some discussion of guidelines, principles, or references to other work explaining how tuning can be done, and some acknowledgement that the meaning of fairness may change dramatically depending on that tuning. * Clarity The paper is well organized and explained. It could be improved by some acknowledgement that there are a number of other (competing, often contradictory) definitions of fairness, and that the two appearing as constraints in the present work can in fact be contradictory in such a way that the optimization problem may be infeasible for some values of the tuning parameters. * Originality The most closely related work of Zemel et al. (2013) is referenced, the present paper explains how it is different, and gives comparisons in simulations. It could be improved by making these comparisons more systematic with respect to the tuning of each method--i.e. compare the best performance of each. * Significance The broad problem addressed here is of the utmost importance. I believe the popularity of (IF) and modularity of using preprocessing to address fairness means the present paper is likely to be used or built upon.

Reviewer 3



The authors propose a dataset model that is the solution of a convex problem, incorporating constraints that limit discrimination w.r.t. a selected sensitive property. The constraints are such that enforce statistical parity both in the joint distribution, as is usual in the literature, but also suggesting quadratically many (in the value outcomes) extra constraints that limit the discriminatory effect per value of the protected attribute. Additional constraints limit individual (point-wise) discrimination, while retaining as close as possible a distribution to the original data. The model can be used to principally limit discrimination from training data, as well as to appropriately transform unseen, test data. The claims are theoretically grounded and what is more in a way that different distance/similarity measures may be incorporated into the constraints. For a selection of these settings, experiments are provided on representative datasets, which demonstrate that their method successfully avoids discrimination, also compared against alternatives in the literature. In the extensive Supplementary Material, proofs are provided that support their theoretical claims, accompanied by insightful discussion about the datasets used (alongside the recently released COMPASS dataset), further compared to the resulting datasets of their algorithm. One further positive aspect is a novel mention of a bound on the utility of their method, in case that the dataset probabilities are not exact but instead estimated using the maximum likelihood estimate, as is usually the case. Unsurprisingly, the tigthness of the bound is subject to sufficiently well-behaving distributions. The authors carefully position their proposal with regard to existing methods in the ecosystem of discrimination-aware ML/DM, and give a good overview of the relevant literature. Overall I am positive of this work, yet there are points that could be improved upon, though, including the following. i) For the distribution similarity (Eq. 3) they choose a not well-motivated measure, that of the probability ratio, which may unevenly penalise differences when the target distribution is very low. ii) The experimentally used distance (distortion) measures are not thoroughly explained (use of 10^4 penalisation in the case of COMPASS data in the case of more than 1 category jumps), although the high value seems reasonable. iii) Experimental results in the Adult dataset are failing behind than the LFR method of Zemer. iv) The experimental section could benefit from more case studies that show a more general trend (e.g. German credit, Heritage Health Prize, etc) v) Value is put in the generality of their framework, which must be solved using a general convex solver. Owing further to the variety of configurations, it becomes difficult to provide reasonably generalisable timings of their proposed optimisation problem. However, it still seems that the submission would be more complete by reporting timing measurements for the provided experiments. Although the reviewer has not thoroughly verified the correctness of the proofs in the supplementary material, the claims made are reasonable, some are novel and all are well motivated, in a way that unifies advances in the area. The reviewer is confident that this submission extends significantly the area of discrimination-aware machine learning, and is confident that the publication of this work would definitely benefit the community.

Reviewer 4



Summary: The authors propose to use a probabilistic transformation of training data as preprocessing to avoid algorithmic discrimination (prejuducial treatment of an individual based on legally protected attributes such as race, gender, etc.) by machine-learned algorithms. The transformation is optimized to produce a new distribution as close as possible to the old distribution while respecting two constraints: -Distortion control: Each individual's representation should not change too much -Discrimination control: The distributions over outcomes, conditioned on protected attributes should be as similar to each other as possible. Overall: The method looks reasonable and technically sound. However, the experiments do not clearly demonstrate an advantage over existing preprocessing work (LFR), nor do they compare to other post-processing or in-processing procedures, so it is difficult to draw a practical conclusion of whether/when it would be useful to use this new method. Detailed comments: Related work / experimental comparisons: - While I understand the appeal of preprocessing over in-processing and post-processing methods, I would still be interested in seeing experimental results that explicitly compare to these other methods as it would be helpful to practically decide whether it is worth investing in more complex procedures. Methods: - Defining a different metric for distortion for every new dataset sounds like it could be difficult. What is the sensitivity of the results to the choices made in determining these measures? - The distortion constraint is currently formulated in expectation, which means that a small number of individuals may experience extreme shifts in features. That sounds like it could be "unfair" to particular individuals who were initially extremely suitable for e.g., a loan application, and then had their features shifted drastically (although it happens with low probability). - Discrimination constraint is formulated in terms of distribution of the labels. The authors recognize that conditioning may be necessary to avoid Simpson's paradox. It would also be worth discussing other notions of fairness - e.g., calibration (see Definition 1 in "Fair prediction with disparate impact" by Chouldechova) which requires conditioning on the predictive score which is a function of all of the features. While the current work may not be compatible with the calibration definition of fairness, I think this would be worth discussing explicitly. - The two constraints in the optimization problem (distortion & discrimination) are at odds with each other. It seems like there may be settings where there is no feasible solution. At that point, the practitioner would then need to either relax their thresholds and try again. This sounds non-ideal. Does this say something about the dataset if it requires too much distortion to become fair? - The authors assume that a small KL-divergence between two the original distribution and the target distribution will maintain the utility of the classifier. Is this provable? Offhand I don't have any obvious counter-examples, but it seems that it may be possible to have a small KL-divergence but a big change in utility between the original and transformed datasets. It would seem that this would also depend on the hypothesis class from which classifiers are being chosen? Results - LFR looks like it dominates the proposed approach on the COMPAS dataset (higher AUC with less discrimination). On the Adult dataset it operates a different operating characteristic so it is hard to compare. It would be useful to see a range of parameters being used in LFR to see the range of its performance on the Adult dataset. Overall, the experimental results do not give a clear picture that the proposed result is better or even comparable to LFR. The proposed method does give an explicit way of setting the discrimination constraint - which is a plus. - The losses in AUC due to the proposed approach seem significant on both data sets (much bigger than e.g., the difference between a log-linear model and random forest) which makes me question the practicality of applying this method in real applications.